# Exploring the Chemical Constituents, Antioxidant, Xanthine Oxidase and COX Inhibitory Activity of *Commiphora gileadensis* Commonly Grown Wild in Saudi Arabia

**DOI:** 10.3390/molecules28052321

**Published:** 2023-03-02

**Authors:** Khalid A. Shadid, Ashok K. Shakya, Rajashri R. Naik, Talal S. Al-Qaisi, Ghaleb A. Oriquat, Ali M. Atoom, Husni S. Farah

**Affiliations:** 1Pharmacological and Diagnostic Research Center, Department of Pharmaceutical Sciences, Faculty of Pharmacy, Al-Ahliyya Amman University, Amman 19328, Jordan; 2Pharmacological and Diagnostic Research Center, Faculty of Allied Medical Sciences, Al-Ahliyya Amman University, Amman 19328, Jordan

**Keywords:** *Commiphora gileadensis*, bisham, essential oil, antioxidant, xanthine oxidase, DPPH, protein denaturation, total phenolic content, total flavonoid content, GC-MS, LC-MS/MS, balm of Makkah

## Abstract

The use of the synthetic drugs has increased in the last few decades; however, these drugs exhibit various side effects. Scientists are therefore seeking alternatives from natural sources. *Commiphora gileadensis* has long been used to treat various disorders. It is commonly known as bisham or balm of Makkah. This plant contains various phytochemicals, including polyphenols and flavonoids, with biological potential. We found that steam-distilled essential oil of *C. gileadensis* exhibited higher antioxidant activity (IC_50,_ 22.2 µg/mL) than ascorbic acid (IC_50,_ 1.25 µg/mL). The major constituents (>2%) in the essential oil were β-myrcene, nonane, verticiol, β-phellandrene, β-cadinene, terpinen-4-ol, β-eudesmol, α-pinene, cis-β-copaene and verticillol, which might be responsible for the antioxidant and antimicrobial activity against Gram-positive bacteria. The extract of *C. gileadensis* exhibited inhibitory activity against cyclooxygenase (IC_50,_ 450.1 µg/mL), xanthine oxidase (251.2 µg/mL) and protein denaturation (110.5 µg/mL) compared to standard treatments, making it a viable treatment from a natural plant source. LC-MS analysis revealed the presence of phenolic compounds such as caffeic acid phenyl ester, hesperetin, hesperidin, chrysin and transient amounts of catechin, gallic acid, rutin and caffeic acid. The chemical constituents of this plant can be explored further to investigate its wide variety of therapeutic potential.

## 1. Introduction

In the last few decades, the use of synthetic drugs has increased massively; however, the use of such products comes with undesired side effects. Scientists are therefore looking for alternatives from natural sources. Since time immemorial the traditional medicine practitioners have been using the mixture of herbs or mixture of plant extracts to treat various disorders [1]. *Commiphora gileadensis* is one such plant. It has been used for centuries to treat various disorders [2,3]. In Arab countries, *C*. *gileadensis* is traditionally known as balm of Makkah, basin and bisham [4]. It belongs to the Burseraceae family, in which there are around 190 species that belong to the genus *Commiphora* distributed in regions such as southern Arabia (Yemen and Oman); in northeastern African countries such as Somalia, Ethiopia, Sudan; and in subcontinental countries such as India and Pakistan [5,6,7]. Its resin is widely used in perfumes and, due to its medicinal properties, it is used in medicinal products to treat various disorders including arthritis, obesity, pain, and gastrointestinal and parasitic infections [7,8]. It is found in the Sarawat mountains in the west of Saudi Arabia. One study used an aqueous extract of *C*. *gileadensis* as an analgesic, diuretic and antihypertensive agent [9]. In another study, an aqueous extract induced anti-inflammatory effects in rats [10,11]. However, few authors have examined the antibacterial activity of the aqueous extract [12,13,14]. In this paper, we report on the chemical constituents and biological activities of an extract of *C*. *gileadensis*, which include antioxidant activity, the inhibition of protein denaturation and xanthine oxidase activity from the aerial parts of the plant. 

## 2. Results

### 2.1. Yield

The yield of ethanolic extract was 37% and the essential oil was 0.26%. GC-MS and LC-MS/MS analyses were conducted to identify its constituents. The results of the GC-MS analysis of the extract and essential oil are presented in Table 1 and Table 2, as well as Appendix A.

### 2.2. Total Phenolics and Flavonoids

The total phenolics and flavonoids in 70% ethanolic extracts of *C. gileadensis* were 18.90 mg GAE/g extract and 59.41 mg QE/g extract, respectively (Table 3). 

### 2.3. GC-MS Analysis

Chemical composition of ethanolic extract and essential oil.

The chemical composition of the ethanolic extract of *C. gileadensis* was analyzed by GC-MS [15]. The percentage yield of the oil was 0.26%. The extract contained monoterpene hydrocarbons (mh, 2.86%), oxygenated mono-terpenes (om, 5.52), sesquiterpenes hydrocarbons (sh, 56.77%), oxygenated sesquiterpenes (os, 12.21), diterpene (dh, 0.05), non terpenes (nt, 2.64%) and transient amounts of oxygenated diterpenes and tetraterpene (th). Around 80.05% of volatile compounds were identified by GC-MS. The data indicate that the essential oil from the aerial part of the plant contained a high percentage of sesquiterpene compounds. Sesquiterpene compounds such as Copaene (11.48%), β-Selinene (5.02%), α-Muurolene (9.01%), τ-Cadinol (3.61%), (Z)-β-Elemene (3.58%), Cubenene (3.48%), α-Cubebene (3.3%), trans-Calamenene (2.65%), γ-Muurolene (2.32%), (+)-spathulenol (1.28%), Cadalene (1.27%), Aromandendrene (1.25%), Ylangene (1.15%), δ-muurolene (1.09%), 1,4-Methanoindan (1.03%), γ-Muurolene (0.97%), δ-EIemene (0.94%), Gurjunene (0.91%), τ-Cadinol (0.54%) and α-Calacorene (0.44%) were identified using a mass library and the injection of standard compounds. 

As regards the essential oil, the freshly extracted steam-distilled oil had more than 118 compounds, of which 91.2% were identified. The percentage of steam-distilled oil was 0.26%. In total, 36.9% monoterpene hydrocarbons, 14.0% diterpene hydrocarbons, 13.7% non-terpenes (nt) and 12.5% sesquiterpene hydrocarbons were detected in the essential oil of *C. gileadensis*. Other chemical constituents detected included oxygenated diterpenes (2.5%), oxygenated monoterpenes (4.5%) and oxygenated sesquiterpenes (7.3%), along with around 8.8% unknown compounds. The major chemical constituents were β-myrcene (17.44%), nonane (10.88%), verticiol (10.56%), β–phellandrene (9.59%), β–cadinene (3.64%) and terpinen-4-ol (3.55%). β–Eudesmol, α-pinene, cis–β-copanene, verticillol were present in the quantities of 2.71%, 2.51%, 2.47% and 2.26%, respectively. Other constituents minor constituents in the essential oil included γ-terpinene (1.79%), trans–β-ocimene (1.68%), thunbergen (1.54%), cembrene (1.52%), epiglobulol (1.43%), trans-calamenene (1.35%) and α-terpinene 1.01%, with unknown constituents forming (8.8%). Chemical constituents present at less than one percent included 7-epi-α-cadinene (0.99%), γ–muurolene (0.84%), linolenic alcohol (0.83%), (-)β-cadinol (0.82%), pseudo limonene (0.73%), α-terpinolene (0.66%), β-bisabolene (0.65%) and linalool (0.52%). 

The percentage of these compounds was low compared to the major compounds identified in monoterpenes. Dudai et al. [16] obtained similar findings regarding a high percentage of monoterpenes compared to sesquiterpenes. 

However, there are other reports that contrast with the present study. The chemical components of the essential oil of the aerial parts and flower of *C. gileadensis*, collected in Makkah, Saudi Arabia, were dominated by sesquiterpenes and the absence of monoterpenes, with the exception of terpinene-4-ol (8.5% and 9.8%) [12]. A similar observation was observed in the essential oil of the stem and bark of *C. gileadensis* collected from Ein-gedi Gardens [17]. In contrast to our investigation, which was carried out on the aerial parts of *C. gileadensis* collected from Saudi Arabia, this variation in the chemical constituents may be due to time of collection, difference in the sample processing, or diversity among specimens of the same plant species.

### 2.4. LC-MS/MS Analysis

The targeted analysis of ethanolic extract of *C. gileadensis* revealed the presence of CAPE, hesperetin, hesperidin and chrysin, as well as small amounts of catechin, gallic acid, rutin and apigenin (Table 4). It also contained transient amounts of caffeic acid and myricetin.

These phenolic compounds are involved in biological activity such as DPPH free radical scavenging, xanthine oxidase inhibition and inhibition of protein denaturation. The ethanolic extract displays moderate antibacterial activity against *S. aureus*. The extract did not exhibit appreciable antibacterial activity against *E. coli* (Figure 1) due to the presence of lipophilic compounds which cannot cross the Gram-negative bacterial cell wall.

## 3. Discussion

### 3.1. Biological Activity

#### 3.1.1. Antioxidant Activity

Antioxidants protect cells from the adverse effect of foreign molecules, drugs, carcinogenic substances and free radicals [18]. Antioxidants can reduce free radicals by scavenging it. Free radicals are generated in various metabolic processes and are associated with a number of stress-related diseases including cancer, diabetes, dementia and necrosis of the myocardial cells due to interaction with DNA, which causes causing mutation [18,19,20]. Extract of *C. gileadensis* showed antioxidant activity due to the presence of various chemical constituents such as polyphenols, flavonoids, catalase and oxidase. Therefore, plant-based products or mixtures that include polyphenols and flavonoids possess antioxidants that can eliminate free radicals and protect human health and wellbeing [21]. In the present study, antioxidant activity was analyzed using DPPH. Free radical scavenging activity was also evaluated using DPPH, and it was found that the ethanolic extract and essential oil showed potent antioxidant activity with IC_50_ values of 56.5 ± 0.4 and 22.2 ± 0.5, respectively, compared to ascorbic acid (1.25 ± 0.05 µg/mL) which is a known antioxidant molecule. The presence of compounds such as α-pinene, *β*-pinene, *γ*-terpinene and terpinen-4-ol may have contributed to the antioxidant activity observed [22]. Ethanolic acid was also evaluated for inhibition using a *β*-carotene bleaching assay. The ethanolic extract showed appreciable activity with an IC_50_ value of 75.8 ± 7.7 µg/mL compared to rutin (4.50 ± 0.35 µg/mL). This might be due to the presence of gallic acid, caffeic acid, rutin, hesperidin and other compounds present in ethanolic extract and the aerial parts of the tree. The phenolic compounds and flavonoids (gallic acid, caffeic acid and rutin) present in the ethanolic extract inhibit the denaturation of protein. The results are presented in the Table 5. The aqueous extracts of *C. gileadensis* leaf and twig have been shown to be successful in treating alloxan-induced diabetes in hypercholesterolemic male rats, where they restored all biochemical parameters to normal [23].

#### 3.1.2. COX-1 Inhibitory Activity

Evaluation of COX activity is the basic strategy for developing NSAIDs to treat inflammation-related disorders. To test the COX-1 inhibitory effect of the ethanolic extract, a COX-1 inhibitory screening assay was performed using commercial assay kits (ab204698, Abcam, Tokyo, Japan) containing SC560 as a standard. The assay was performed as per the instruction provided with the kit. Different concentrations of ethanolic extract in 5% DMSO were used, with 5% DMSO used as a control. The preliminary screening results indicated that the ethanolic extract of *C. gileadensis* displayed COX-1 inhibitory activity at a concentration of 450 µg/mL. Under comparable conditions, the IC_50_ of the standard compound SC560 was 5 ng/mL. It is possible that the CAPE, caffeic acid, gallic acid and other phenolic compounds are responsible for COX-1 inhibitory activity. More investigations would be required to evaluate COX-inhibitory activity on the fractionated ethanolic extract. Furthermore, NSAIDs’ primary mode of action prior to Vane’s discovery [24] of their inhibitory activity against cyclooxygenase is the suppression of protein denaturation [25]. Ben-Yehoshua et al. [26] have reviewed the COX-2 inhibitory activity of chloroform extract of *C. gileadensis.*

#### 3.1.3. Xanthine Oxidase Activity

One of main characteristics of the ischemic injury is the overproduction of superoxide anion and the conversion of xanthine dehydrogenase to xanthine oxidase, which produces superoxide anions when xanthine is converted to uric acid [27]. Some natural products containing polyphenols have shown a dose-dependent inhibitory effect [28]. The flavonoids and phenolic acids present in plant products are potent inhibitors of enzymes such as cyclooxygenase, xanthine oxidase and lipo-oxygenase [29]. These enzymes control inflammation, hyperuricemia and gout. The chemical constituents gallic acid, rutin, caffeic acid and CAPE play important roles in the inhibition of xanthine oxidase. Inhibitors of xanthine oxidase and uricosuric agents are used in the treatment of diseases such as gouty arthritis and inflammatory diseases. At present, drugs such as allopurinol are used in the treatment of gout, but these synthetic drugs have various side effects [30]. Drugs with greater therapeutic values and lesser side effects are required. As the ethanolic extract of *C. gileadensis* contains flavonoids and phenolic compounds like rutin, apigenin, catechin, hesperetin, hesperidin, CAPE and caffeic acid, the xanthine oxidase activity of its ethanolic extract was investigated.

A literature survey on xanthine oxidase activity revealed no publications on the inhibitory activity of *C. gileadensis* in the treatment of xanthine oxidase and gout. The oil extracted from the aerial part of the plant exhibited significant xanthine oxidase activity. Inhibitory activity data is expressed as inhibitory concentration (µg/mL, Table 5). Wang et al. [31] reported that the caffeic acid, its derivatives and other flavonoids can inhibit xanthine oxidase.

## 4. Materials and Methods

### 4.1. Collection of Plant Material and Sample Preparation

The sample was collected in spring 2019 from the Badr area in Medina district, Al-Hijaz, Saudi Arabia. The aerial parts of the plant were washed, dried and ground for sample processing (Figure 2).

#### 4.1.1. Ethanolic Extract of the Aerial Part of *C. gileadensis*

The ground aerial parts (0.25 kg) were soaked in 50% ethanol (2.5 L) with occasional shaking in two different containers for two days. The content was warmed briefly (60 °C) for 5 h and then filtered under a vacuum using a sintered glass funnel G2. The filtrate was evaporated collectively using a Buchi R-100 Rotary Evaporator (BÜCHI Labortechnik AG, Flawil, Switzerland). The samples were then stored in liquid nitrogen for analysis.

#### 4.1.2. Extraction of Essential Oil

The essential oil of *C. gileadensis* was extracted from 100 g of the prepared sample in 500 mL of water by hydro-distillation for four hours in a Clevenger-type apparatus. After extraction, the essential oil was dried over anhydrous sodium sulfate and stored in light-resistant, inert glass tubes. The essential oil was kept at 2–8 °C until further investigation.

### 4.2. Phytochemical Analysis

#### 4.2.1. Total Flavonoid Content

Using the technique outlined by Al-Jaber et al. [32], the total flavonoid content in the ethanolic extract was determined calorimetrically using the reported technique without any modification. The flavonoid concentration was expressed as mg quercetin equivalent/g of dry extract (mg QE/g).

#### 4.2.2. Total Phenolic Content

The Folin–Ciocalteu method was used for measuring the total phenol content in the extract. Briefly, a mixture of 2.5 mL Folin–Ciocalteu reagent (2 N diluted tenfold) and 2 mL of Na_2_CO_3_ solution were mixed with 0.5 mL of the extract. The resultant mixture was allowed to stand at room temperature for 15 min. The absorbance of the solution was determined at 765 nm using methanol as the blank solution. Then, total phenol content was given as mg gallic acid equivalent/g of dry extract (mg GAE/g). Each measurement was carried out three times [32].

### 4.3. GC-MS Analysis

The procedure outlined by Halub et al. and Naik et al. [15,33] using a Shimadzu QP2020 GC-MS (Shimadzu Corporation, Kyoto, Japan) supplied with a split-split less injector, was utilized to analyze samples and separate the volatile components using a DB5-MS fused silica column. Besides the published data, every chemical component’s mass spectrum was compared to the corresponding reported spectra for GC-MS, with reference to the ADAMS-2007 and NIST 2017 mass spectrometry libraries. To ascertain the identified compound, a comparison was made between reported values and relative retention indices (RRI) in reference to n-alkanes (C_8_–C_30_).

### 4.4. LC-MS/MS Analysis of the Extract

Analysis of flavonoids and phenolics compounds was performed using a SciEx UPLC (Exion-UPLC, USA) equipped with the LC-ESI-PDA-MS/MS-4500-QTRAP system (AB Sciex Instrument, Framingham, MA, USA), utilizing Analyst 1.7 software for data analysis. The chromatographic separation was conducted at 30 ± 1 °C using a Phenomenex column (3.0 × 50 mm, 5 µm). The elution gradient consisted of a mobile phase A (5 mM ammonium format in water: methanol (95:5; *v*/*v*)) and methanol (1 mM formic acid). The gradient program with the following proportions of solvent B was applied (%B, min): 5–90% B (0.00–8.00 min), 90–90% (8.00–12.00 min), 90–5% (12.01–15.00 min). The solvent flow rate was 0.35 mL/min and the injection volume was 5 µL. MS/MS analysis was performed in positive and negative ion mode. Nitrogen gas at a pressure of 60 psi was used as the nebulizing and drying gas. The mass spectra were obtained over an m/z range of 100–900 amu.

### 4.5. In Vitro Antioxidant Studies

#### 4.5.1. DPPH (2-Diphenyl-1-Picryl-Hydrazyl) Free Radical Scavenging Activity

According to Tulini et al. [33], measuring antioxidant activity involves establishing the amount of antioxidant needed to decrease the initial DPPH concentration to 50% (IC_50_). A DPPH stock solution (0.002 percent *w*/*v*) in ethanol was prepared. As a serial dilution process, various ethanoic concentrations of the extract or essential oil (4–500 μg/mL) were prepared. DPPH solution (150 μL) was mixed in an ELISA plate with 150 μL of the sample at various concentrations. ELISA plates were incubated in the dark for 30 min until the color developed. Using a multi-mode ELISA reader (Synergy HTX, Biotek, CA, USA) the developed color was measured at 517 nm. DPPH radical scavenging activity was determined and the IC_50_ was calculated using SigmaPlot ver. 11 [34].
(1)$$\%\;\mathrm{Free}\;\mathrm{radical}\;\mathrm{Scavenging}\;\mathrm{activity}=\frac{\left[\mathrm{Abs}.\mathrm{control}-\mathrm{Abs}.\mathrm{sample}\right]}{\mathrm{Abs}.\mathrm{control}}\times 100$$

#### 4.5.2. *β*-Carotene Bleaching (BCB) Assay

A solution of *β*-carotene/linolenic acid was prepared by dissolving 10 mg of *β*-carotene as a stock solution into 100 mL of chloroform. In another flask, linolenic acid (25 mg) and Tween 20 (215 mg) were mixed and 3 mL of *β*-carotene solution was then added. The chloroform was evaporated by flushing with nitrogen gas. Next 75 ml of distilled water was added to the *β*-carotene/linolenic acid solution. Immediately after preparation, the absorbance of this solution was recorded at 470 and 700 nm. A standard antioxidant (rutin) solution was prepared by dissolving 3 mg of rutin in 10 mL methanol (300 μg/mL) as stock, and then various concentrations of rutin were prepared (0.6–19 μg/mL). Different strength solutions of extract (15–500 μg/mL) were prepared in methanol in an ELISA plate; 275 μL of *β*-carotene/linolenic acid solution was added to all the cells, each of which contained 25 μL of extract, methanol and rutin standard. All the solutions (control and test) were incubated (50 °C) for 1 h. The absorbance was taken at 470 nm and 700 nm at zero time, then after 1 h and 2 h. The control sample contained the equivalent amount of methanol. The degradation rate and antioxidant activity were calculated using the formula:Degradation rate of *β*-carotene = Ln (A initial/A Sample)/60(2)
(3)$$\mathrm{Antioxidant}\;\mathrm{activity}\;(100\%)=\frac{\mathrm{Degradation}\;\mathrm{rate}\;\mathrm{of}\;\mathrm{control}-\mathrm{degradation}\;\mathrm{rate}\;\mathrm{of}\;\mathrm{sample}}{\mathrm{degradation}\;\mathrm{rate}\;\mathrm{of}\;\mathrm{sample}}\times 100$$

### 4.6. COX-1 Inhibitory Activity

COX-1 inhibitory activity was measured using a COX-1 kit (ab204698, Abcam, Japan) as per the instructions provided by the supplier [35]. The samples were mixed with a cofactor, COX-1 enzyme and arachidonic acid. The enzymatic reaction was stopped by adding a 0.2 mL aliquot of 1 N HCl. The positive control for the reaction was SC-560.

### 4.7. Xanthine Oxidase Inhibitory Activity

Xanthine oxidase (XO) inhibitory activity was measured by observing the formation of uric acid in the xanthine oxidase system, as described in [36]. The assay kit consisted of 0.6 mL phosphate buffer (100 mM; pH 7.4), 0.1 mL sample, 0.1 mL XO (0.2 U/mL) and 0.2 mL xanthine (1 mM; dissolved in 0.1 N NaOH). For the analysis, the reaction was initialized by the addition of enzymes with or without inhibitors. Changes in the absorbance of the reaction mixture at 290 nm for 15 min were determined by comparing it with the absorbance of the reagent blank. The enzymatic reaction was stopped by adding a 0.2 mL aliquot of 1 N HCl. The positive control for the reaction was allopurinol.

### 4.8. Inhibition of Protein Denaturation

Inhibition of protein denaturation was estimated using the reported procedure [37] with minor modifications. Various solutions for the assay procedure were prepared: test solution, test control, product control and standard solution. All the required solutions were prepared using a pH 6.3 buffer. The samples were incubated for 20 min at 37 °C, then the temperature was raised to 50 °C and they were incubated for a further 15 min. After cooling, the absorbance was determined at 416 nm with a Synergy HTC multimode reader (BioTek, Winooski, VT, USA). The percent inhibition of protein denaturation was calculated using the given formula.

### 4.9. Antimicrobial Activity

Gram-positive (*S. aureus* ATCC 6538) and Gram-negative (*E. coli* ATCC 8739) bacteria were used to assess the ability of the extract and moxifloxacin (standard drug) to inhibit bacterial growth [38]. Sterilized nutrient agar was placed in a 9 cm sterilized petri disc. The plates were inoculated with bacterial culture using sterilized cotton swaps. Next, 6 mm holes were created with sterilized 6 mm surgical punches. The wells were filled with 40 µL of extract solution (250 µg/mL), essential oil and moxifloxacin standard solution. A 5% DMSO solution was used as a control. The plates were incubated at 37 °C for 24 h. Each experiment was performed in duplicate. Zones of inhibition were measured and are reported in Figure 1 and Table 6.

#### Statistical Analysis

The results obtained in the present study are expressed as mean ± standard deviation (SD). For the statistical analysis of the experimental data, Graph-Pad Prism 5 (Graph-Pad Software, San Diego, CA, USA) was used.

## 5. Conclusions

It can be concluded that due to the presence of various chemical constituents such as gallic acid, rutin, apigenin, caryophyllene, α-pinene, *β*-pinene, *γ*-terpinene and terpinen-4-ol, the *C*. *gileadensis* extract exhibited significant biological activity. The various chemical constituents present in the plant extract make it a viable natural resource that can be used in various pharmacological formulations. The plant extract showed significant biological activity, which contributes to its success as a traditional medicine used by traditional practitioners to treat various ailments including inflammation, pain and gouty arthritis. The chemical components can be explored further for their various biological and pharmacological benefits.

## Figures and Tables

**Figure 1 molecules-28-02321-f001:**
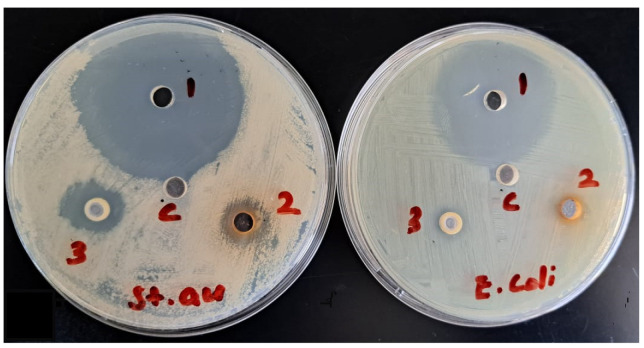
Antibacterial activity of essential oil (spot 3); ethanolic extract (spot 2); and moxifloxacin (spot 1) against *S. aureus* and *E. coli.* Control (5% DMSO) is marked “C”.

**Figure 2 molecules-28-02321-f002:**
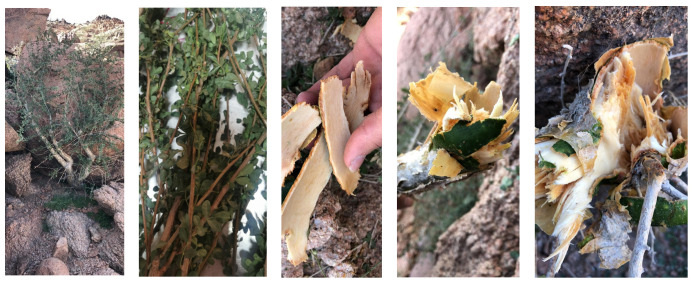
Photos of the plant, harvested in Badr, Medina district, Saudi Arabia.

**Table 1 molecules-28-02321-t001:** GC-MS analysis of the ethanolic extract of *C. gileadensis*.

No.	Class	CAS Registry Number	Name	Area %	Ret. Index
1	mh	2867-05-2	α-Thujene	0.15	929
2	nt	18829-55-5	(E)-2-Hepten-1-al	0.02	958
3	mh	555-10-2	β-phellandrene	0.60	976
4	nt	110-93-0	6-Methyl-5-hepten-2-one	0.07	984
5	mh	123-35-3	β-Myrcene	0.02	991
6	nt	123-66-0	Ethyl caproate	0.01	999
7	mh	99-86-5	α-Terpinene	0.09	1020
8	mh	527-84-4	o-Cymene	0.61	1027
9	om	470-82-6	Eucalyptol	0.44	1035
10	mh	99-85-4	γ-Terpinene	0.45	1060
11	om	138-87-4	β-Terpineol	0.54	1073
12	om	1365-19-1	Linalool oxide	0.25	1101
13	om	546-79-2	4-Thujanol	0.77	1104
14	nt	60-12-8	Phenylethyl alcohol	0.23	1114
15	om	471-15-8	β-Thujone	0.04	1021
16	om	546-79-2	4-Thujanol	0.81	1131
17	om	432-25-7	β-Cyclocitral	0.01	1130
18	om	471-16-9	Cis-Sabinol	0.02	1142
19	om	1674-08-4	trans-Pinocarveol	0.10	1145
20	om	1845-30-3	cis-Verbenol	0.06	1148
21	om	74410-10-9	Anethofuran	0.04	1152
22	om	17699-16-0	trans-4-Thujanol	0.81	1161
23	om	30460-92-5	Pinocarvone	0.06	1166
24	nt	7786-44-9	2,6-Nonadien-1-ol	0.08	1172
25	om	33522-69-9	trans-Verbenyl acetate	0.32	1179
26	mh	562-74-3	Terpinen-4-ol	0.94	1185
27	nt	1197-01-9	p-Cymen-8-ol	0.19	1189
28	nt	106-32-1	Ethyl octanoate	0.09	1195
29	om	32543-51-4	cis-Limonene oxide	0.41	1200
30	nt	112-31-2	Decanal	0.05	1206
31	om	18881-04-4	cis-Verbenone	0.08	1212
32	nt	10519-33-2	3-Decen-2-one	0.05	1215
33	om	1197-06-4	cis-Carveol	0.04	1221
34	nt	101-97-3	Benzeneacetic acid, ethyl ester	0.02	1243
35	om	122-03-2	p-Cumic aldehyde	0.06	1246
36	om	115-95-7	Linalyl acetate	0.12	1249
37	om	20777-49-5	Dihydrocarvyl acetate	0.14	1252
38	nt	3913-81-3	2(E)-Decenal	0.05	1263
39	om	13040-03-4	cis-Verbenol	0.05	1282
40	om	76-49-3	Bornyl acetate	0.18	1287
41	om	73366-12-8	(+)-cis-Verbenol, acetate	0.03	1290
42	nt	123-29-5	Nonanoic acid, ethyl ester	0.17	1294
43	om	1619-26-7	trans-ascaridol	0.14	1308
44	sh	20307-84-0	δ-Elemene	0.94	1339
45	sh	17699-14-8	α-Cubebene	3.30	1353
46	sh	14912-44-8	Ylangene	1.15	1376
47	sh	3856-25-5	Copaene	11.48	1389
48	sh	33880-83-0	(Z)-β-Elemene	3.58	1399
49	sh	6813-05-4	Sativene	0.23	1407
50	sh	67650-50-4	Gurjunene	0.91	1411
51	sh	87-44-5	Caryophyllene	0.53	1429
52	nt	581-42-0	2,6-Dimethylnaphthalene	0.09	1431
53	sh	11028-42-5	Cedrene	0.27	1435
54	sh	120021-96-7	δ-muurolene	1.09	1438
55	sh	3650-28-0	1,4-Methanoindan	1.03	1444
56	sh	489-39-4	Aromandendrene	1.25	1447
57	sh	23986-74-5	Germacrene D	0.41	1452
58	sh	95910-36-4	isoledene	0.29	1456
59	sh	20085-19-2	Amorphene	0.19	1458
60	sh	6753-98-6	Humulene	0.39	1464
61	sh	30021-74-0	γ-Muurolene	0.97	1468
62	nt	18435-22-8	3-methyltetradecane	0.06	1473
63	sh	18431-82-8	Chamigren	0.12	1491
64	nt	13360-61-7	1-Pentadecene	0.87	1496
65	sh	17066-67-0	β-Selinene	5.02	1501
66	sh	10208-80-7	α -Muurolene	9.01	1508
67	sh	30021-74-0	γ-Muurolene	2.32	1522
68	sh	29837-12-5	Cubenene	3.48	1527
69	sh	73209-42-4	trans-Calamenene	2.65	1532
70	os	77-53-2	Cedrol	3.40	1555
71	sh	5937-11-1	τ-Cadinol	3.61	1568
72	sh	21391-99-1	α-Calacorene	0.44	1571
73	os	28231-03-0	cedr-8(15)-en-9-α-ol	0.43	1574
74	nt	6789-88-4	Benzoic acid, hexyl ester	0.46	1583
75	os	6750-60-3	(+)-spathulenol	1.28	1587
76	os	515-69-5	α-Bisabolol	1.47	1593
77	sh	28908-27-2	β-Vetispirene	0.12	1597
78	os	489-41-8	Globulol	2.46	1606
79	os	577-27-5	Ledol	0.34	1615
80	os	1139-30-6	Caryophyllene oxide	0.13	1620
81	os	28231-03-0	Cedrenol	0.80	1634
82	os	19912-62-0	τ-Muurolol	0.79	1636
83	sh	5937-11-1	τ-Cadinol	0.54	1651
84	os	28387-62-4	α-Cedr-8(15)-en-9-al	0.45	1655
85	os	88728-58-9	Epiglobulol	0.61	1676
86	sh	483-78-3	Cadalene	1.27	1681
87	os	77171-55-2	(-)-Spathulenol	0.04	1703
88	nt	544-63-8	Tetradecanoic acid	0.03	1754
89	os	502-69-2	Hexahydrofarnesyl acetone	0.01	1839
90	dh	64363-64-0	Cembrene C	0.03	1932
91	nt	57-10-3	n-Hexadecanoic acid	0.09	1955
92	dh	71213-92-8	3-(Z)-Cembrene A	0.01	1974
93	dh	70000-19-0	Verticiol	0.01	2027
94	od	25269-17-4	Thunbergol	tr	2241
95	nt	117-81-7	Bis(2-ethylhexyl) phthalate	tr	2527
96	th	7683-64-9	Supraene	tr	2808
			monoterpene hydrocarbons (mh)	2.86	
			oxygenated monoterpenes (om)	5.52	
			sesquiterpenes hydrocarbons (sh)	56.77	
			oxygenated sesquiterpenes (os)	12.21	
			diterpene (dh)	0.05	
			non-terpenes (nt)	2.64	
			oxygenated diterpenes (od)	tr	
			tetraterpene (th)	tr	
			Total identified	80.05	

**Table 2 molecules-28-02321-t002:** GC-MS analysis of the essential oil of aerial parts of *C. gileadensis*.

No.	Class	CAS Registry Number	Name	Area %	Ret. Index
1	nt	505-57-7	2-Hexenal	0.05	850
2	nt	111-84-2	Nonane	10.88	900
3	unknown	-	unknown	0.01	-
4	mh	80-56-8	α-Pinene	2.51	939
5	mh	79-92-5	Camphene	0.02	960
6	mh	555-10-2	β-Phellandrene	9.59	976
7	mh	18172-67-3	β-Pinene	0.22	980
8	mh	123-35-3	β-Myrcene	17.44	992
9	nt	4057-42-5	2,6-dimethyl-oct-2-ene	0.01	-
10	mh	99-83-2	α-Phellandrene	0.16	1007
11	mh	13466-78-9	3-Carene	0.02	1012
12	mh	99-86-5	α-Terpinene	1.01	1020
13	mh	535-77-3	m-Cymene	0.39	1042
14	mh	5989-27-5	D-Limonene	0.43	1031
15	mh	499-97-8	Pseudo limonene	0.73	1033
16	mh	3779-61-1	trans-β-Ocimene	1.68	1041
17	mh	3779-61-1	trans-β-Ocimene	0.29	1046
18	mh	99-85-4	γ-Terpinene	1.79	1060
19	nt	4485-09-0	4-Nonanone	0.05	1030
20	mh	586-62-9	α-Terpinolene	0.66	1063
21	unknown		unknown	0.05	-
22	om	78-70-6	Linalool	0.52	1080
23	om	29803-82-5	cis-p-Mentha-2-en-1-ol	0.20	1123
24	unknown	-	unknown	0.11	-
25	om	562-74-3	Terpinen-4-ol	3.55	1185
26	om	98-55-5	α-Terpineol	0.25	1190
27	nt	112-31-2	Decanal	0.31	1206
28	nt	53398-85-9	cis-3-Hexenyl-α-methylbutyrate	0.06	1235
29	sh	20307-84-0	δ-Elemene	0.05	1339
30	sh	17699-14-8	α-Cubebene	0.08	1353
31	Unknown	-	unknown	0.02	-
32	Unknown	-	unknown	0.90	-
33	sh	5208-59-3	(-)-β-Bourbonene	0.06	1382
34	sh	30021-74-0	γ-Muurolene	0.84	1477
35	sh	489-40-7	α-Gurgujene	0.09	1408
36	sh	87-44-5	Caryophyllene	0.24	1429
37	sh	3691-12-1	α-Guaiene	0.49	1439
38	Unknown	-	unknown	0.02	-
39	sh	3691-11-0	Guaia-1(10),11-diene	0.08	1505
40	Unknown	-	unknown	0.1	-
41	sh	25246-27-9	Alloaromadendrene	0.05	1465
42	sh	523-47-7	β-Cadinene	0.17	1519
43	sh	483-75-0	7-epi-α-Cadinene	0.03	1490
44	sh	18252-44-3	cis-β-Copaene	2.47	1433
45	sh	150320-52-8	2-isopropyl-5-methyl-9-methylene- bicyclo[4.4.0]dec-1-ene	0.23	1503
46	sh	483-75-0	7-epi-α-Cadinene	0.99	1495
47	sh	483-76-1	δ-Cadinene	0.08	1541
48	sh	515-13-9	levo-β-Elemene	0.04	1394
49	sh	39029-41-9	γ-Cadinene	0.11	1512
50	sh	523-47-7	β-Cadinene	3.64	1518
51	sh	73209-42-4	trans-Calamenene	1.35	1532
52	sh	16728-99-7	Cada-1,4-diene	0.12	1532
53	sh	24406-05-1	α-Cadinene	0.03	1538
54	sh	21391-99-1	α-Calacorene	0.34	1548
55	sh	29873-99-2	γ-Elemene	0.16	1434
56	nt	25152-85-6	3-Hexen-1-ol, benzoate, (Z)-	0.41	1570
57	os	198991-79-6	Germacrene -D-4ol	0.30	1574
58	os	515-69-5	α-Bisabolol	0.18	1682
59	os	5937-11-1	*τ*-cadinol	1.60	1640
60	os	88728-58-9	Epiglobulol	1.43	1585
61	os	5986-49-2	Palustrol	0.03	1550
62	os	1139-30-6	Caryophyllene oxide	0.03	1589
63	os	198991-79-6	D-Germacren-4-ol	0.06	1569
64	os	481-34-5	α-Cadinol	0.82	1639
65	os	1209-71-8	γ-Eudesmol	0.03	1597
66	Unknown	-	unknown	0.02	-
67	Unknown	-	unknown	1.66	-
68	os	473-15-4	β-Eudesmol	2.71	1645
69	os	473-04-1	Eudesm-7(11)-en-4-ol	0.14	1689
70	sh	483-78-3	cadalene	0.08	1681
71	Unknown	-	unknown	0.03	-
72	Unknown	-	unknown	0.04	-
73	Unknown	-	unknown	0.24	-
74	Unknown	-	unknown	1.34	-
75	nt	2765-11-9	Pentadecanal	0.35	1707
76	Unknown	-	unknown	0.23	-
77	Unknown	-	unknown	0.45	-
78	Unknown	-	unknown	0.28	-
79	sh	10579-93-8	4,11,11-trimethyl-8-methylene-bicyclo[7.2.0]undec-4-ene	0.04	-
80	nt	506-43-4	(9Z,12Z)-octadeca-9,12-dien-1-ol	0.02	2052
81	nt	506-44-5	Linolenic alcohol	0.83	2058
82	Unknown	-	unknown	0.05	-
83	nt	57-10-3	n-Hexadecanoic acid	0.21	1955
84	Unknown	-	unknown	0.36	-
85	Unknown	-	unknown	0.35	-
86	sh	495-61-4	β-Bisabolene	0.65	1509
87	dh	1898-13-1	Cembrene	1.52	1947
88	mh	500-00-5	p-Menth-3-ene	0.06	1324
89	od	70000-19-0	Verticillol	2.26	2037
90	od	25269-17-4	Thunbergol	0.18	2055
91	Unknown	-	unknown	0.07	1573
92	Unknown	-	unknown	0.51	-
93	od	150-86-7	phytol	0.07	2105
94	Unknown	-	unknown	0.17	-
95	Unknown	-	unknown	0.08	-
96	Unknown	-	unknown	0.79	-
97	Unknown	-	unknown	0.04	-
98	Unknown	-	unknown	0.09	-
99	Unknown	-	unknown	0.06	-
100	Unknown	-	unknown	0.04	-
101	dh	1898-13-1	Thunbergen	1.54	1948
102	dh	70000-19-0	Verticiol	10.56	2027
103	Unknown	-	unknown	0.63	-
104	nt	629-94-7	C-21	0.05	2100
105	Unknown	-	unknown	0.06	-
106	nt	646-31-1	tetracosane C-24	0.1	2400
107	nt	629-99-2	pentacosane C-25	0.08	2500
108	nt	630-01-3	c-26	0.09	2600
109	nt	593-49-7	c-27	0.12	2700
110	nt	630-02-4	c-28	0.06	2800
111	nt	630-03-5	c-29	0.04	2900
	mh		monoterpene hydrocarbons (mh)	36.9%	
	dh		diterpene (dh)	14.0%	
	nt		non-terpenes (nt)	13.7%	
	od		oxygenated diterpenes (od)	2.5%	
	om		oxygenated monoterpenes (om)	4.5%	
	os		oxygenated sesquiterpenes (os)	7.3%	
	sh		sesquiterpenes hydrocarbons (sh)	12.5%	
			unknown	8.8%	
			Total identified	100.0%	

**Table 3 molecules-28-02321-t003:** Total phenolics and flavonoids in ethanolic extracts of *C. gileadensis*.

Sample	Total Phenolics (mg GAE/g Extract)	Total Flavonoids (mg QE/g Extract)
Ethanolic extract (70%)	18.90 ± 0.14	59.41 ± 1.73

Data is given as mean ± SD, *n* = 3.

**Table 4 molecules-28-02321-t004:** LC-MS/MS analysis of the ethanolic extract of *C. gileadensis*.

No.	R_t_ (Minutes)	Name	µg/g of Extract
1	0.14	Gallic acid	100
2	3.12	Hesperetin	4640
3	3.22	Hesperidin	2940
4	4.20	Apigenin	61.6
5	4.48	Chrysin	1080
6	5.96	Rutin	62.5
7	7.01	Caffeic acid phenethyl ester (CAPE)	10,500
8	10.50	Catechin	385
9	10.53	Caffeic acid	Tr
10	10.59	Myricetin	tr

**Table 5 molecules-28-02321-t005:** In vitro DPPH radical scavenging, BCB assay, XO activity and inhibition of protein denaturation activity.

Sample	IC50 (µg/mL)				
	DPPH Radical Activity *	BCB	COX-1 Activity *	XO Activity *	Inhibition of Protein Denaturation
Essential oil (*C. gileadensis*)	22.2 ± 0.5	-	-	-	
Ethanolic extract	56.5 ± 0.4	75.8 ± 7.7	450.1 ± 8.2	251.2 ± 5.6	110.5 ± 5.8
Ascorbic Acid	1.25 ± 0.05	-	-	-	-
Rutin	-	4.5 ± 0.35	-	-	-
SC560	-	-	0.0051 ± 0.0001	-	-
Allopurinol	-	-	-	0.41 ± 0.05	-
Diclofenac Potassium	-	-	-	-	52.2 ± 6.5

* (*n* = 3), IC_50_ (µg/mL) expressed as mean ± sd.

**Table 6 molecules-28-02321-t006:** Antibacterial activity of the essential oil and ethanolic extract against *E. coli* and *Staphylococcus aureus*.

Sample Name	Inhibition Zone (mm)	
	*E. coli*	*S. aureus*
Essential oil (50% emulsion in 5% DMSO, spot 3)	10	20
Ethanolic extract (100 µg/mL 5% DMSO, spot 2)	9	15
Moxifloxacin (25 µg/mL, spot 1)	40	45

*n* = two measurements, diameter of well = 6 mm.

## Data Availability

Data will be provided upon request.

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
