# Peer review of "Exploring the Chemical Constituents, Antioxidant, Xanthine Oxidase and COX Inhibitory Activity of Commiphora gileadensis Commonly Grown Wild in Saudi Arabia"

_molecules, 2023, doi:10.3390/molecules28052321_

Round 1

Reviewer 1 Report

The manuscript of the experimental work entitled "Exploring the chemical constituents, antioxidant, xanthine oxidase and COX inhibitory activity of Commiphora gileadensis commonly grown wild in Saudi Arabia" authors Khalid A. Shadid, Ashok K. Shakya et al is generally well presented. The theme concerning isolation and characterization of an essential oil of plant Commiphora gileadens were clearly described and critically discussed and should be interesting for readers of Molecules. I recommend only adding the Chemical Abstract Service Registry Number (CAS RN) to Table 1 for all compounds where this number has been assigned. The present manuscript should be acceptable after minor revision for publishing in the Molecules.

Author Response

Thank you for reviewing the manuscript. The CAS numbers are updated in table 1 and table 2.

Sincerely Yours,

Ashok

Reviewer 2 Report

In this manuscript, the chemical constituents of the plant Commiphora gileadensis were reported and biological activities were evaluated.

It is a challenging task to analyze the chemical composition in natural plants, this manuscript has analyzed the chemical components in Commiphora gilladensis in detail and carried out the activity research, providing an important reference for studying its extensive therapeutic potential, it is recommended to publish after major revisions.

1.In line 119,“The extract did 119 not exhibit appreciable antibacterial activity against E. coli, which might be due to the presence of chemical compounds having nonpolar characteristics”, references need to be cited to be more convincing.

2. There is no relevant explanation of BCB assessment and inhibition of protein characterization in Table 5. Please complete it.

3. The reason why ethanol extract has COX-1 inhibitory activity is suggested to be guessed or explained.

4.The xanthine oxidation activity of the ethanolic extract” in line 173 needs to be further explained.

5. Please check whether the statement “The results are presented in Table 2.”in line 178 is correct, no data related to xanthine oxidation activity was found in Table 2.

6. The title of the reference only needs the initial letter of the first word, please check carefully.

Author Response

Thank you for comments. Sorry for delay due to travelling and low accessibility to the data. Please find enclosed herewith the reply to the comments. I hope the reviewer will agree with the answers.

With Regards

Ashok

Reviewer 3 Report

In this manuscript the authors extracted in ethanol the Commiphora gileadensis; Bisham. This extract included many important organic compounds, they exhibited the antioxidant and antibacterial activity. The physicochemical methods e.a. GC-MS, LC-MS/MS allowed  to determine the content of flavonoids polyphenols e.t.c.
This manuscript is very well written but the results involving for example antioxidant activity, are lower as ascorbic acid.
The authors should corrected the conclusions because it is to general.

Author Response

Thank you for reviewing and comments, it is revised in the updated copy

Round 2

Reviewer 2 Report

Accept in present form.